# PROBABILISTIC IMPUTATION FOR TIME-SERIES CLASSIFICATION WITH MISSING DATA

## ABSTRACT

Multivariate time series data available for real-world applications typically contain a significant amount of missing values. A dominant approach for the classification with such missing values is to heuristically impute the missing values with specific values (zero, mean, values of adjacent time-steps) or learnable parameters. However, these simple strategies do not take the data generative process into account, and more importantly, do not effectively capture the uncertainty in prediction due to the multiple possibilities for the missing values. In this paper, we propose a novel probabilistic framework for classification with multivariate time series data with missing values. Our model consists of two parts; a deep generative model for missing value imputation and a classifier. Extending the existing deep generative models to better capture structures of time-series data, our deep generative model part is trained to impute the missing values in multiple plausible ways, effectively modeling the uncertainty of the imputation. The classifier part takes the time series data along with the imputed missing values and classifies signals, and is trained to capture the predictive uncertainty due to the multiple possibilities of imputations. Importantly, we show that naïvely combining the generative model and the classifier could result in trivial solutions where the generative model does not produce meaningful imputations. To resolve this, we present a novel regularization technique that can promote the model to produce useful imputation values that actually help classification. Through extensive experiments on real-world time series data with missing values, we demonstrate the effectiveness of our method.

## 1 INTRODUCTION

Multivariate time-series data are universal; many real-world applications ranging from healthcare, stock markets, and weather forecasting take multivariate time-series data as inputs. Arguably the biggest challenge in dealing with such data is the presence of missing values, due to the fundamental difficulty of faithfully measuring data for all time steps. The degree of missing is often severe, so in some applications, more than 90% of data are missing for some features. Therefore, developing an algorithm that can accurately and robustly perform predictions with missing data is considered an important problem to be tackled.

In this paper, we focus on the task of classification, where the primary goal is to classify given multivariate time-series data with missing values, simply imputing the missing values with heuristically chosen values considered to be strong baselines that are often competitive or even better than more sophisticated methods. For instance, one can fill all the missing values with zero, the mean of the data, or values from the previous time steps. GRU-D (Che et al., 2018) proposes a more elaborated imputation algorithm where the missing values are filled with a mixture between the data means and values from the previous time steps with the mixing coefficients learned from the data. While these simple imputation-based methods work surprisingly well (Che et al., 2018; Du et al., 2022), they lack a fundamental mechanism to recover the missing values, especially the underlying generative process of the given time series data.

Dealing with missing data is deeply connected to handling uncertainties originating from the fact that there may be multiple plausible options for filling in the missing values, so it is natural to analyze them with the probabilistic framework. There have been rich literature on statistical analysis for missing data, where the primary goal is to understand how the observed and missing data are

generated. In the seminal work of Little and Rubin (2002), three assumptions for the missing data generative process were introduced, including Missing Completely At Random (MCAR), Missing At Random (MAR), and Missing Not At Random (MNAR). While MCAR or MAR simplifies the modeling and thus makes the inference easier, they may be unrealistic for real-world applications, because they assume that the missing mechanism is independent of the missing values (MAR) or both missing and observed values (MCAR). MNAR, the most generic assumption, assumes that the missing mechanism depends on both missing and observed values, so the generative model based on the MNAR assumption should explicitly take the missing mechanism into account. Based on this framework, Mattei and Frellsen (2019) presented deep generative models for missing data under MAR assumption, and this was later extended to MNAR in Ipsen et al. (2021). Combining the deep generative model and classifier, Ipsen et al. (2022) proposed a hybrid model that can classify missing data with problematically imputed values generated under MAR assumption.

Still, in our opinion, there is no satisfactory work combining probabilistic generative models for multivariate time-series data with missing values and classification models, so that the classifier could consider the uncertainty in filling in the missing values when making predictions. The afore-mentioned probabilistic frameworks are not designed for classification (Mattei and Frellsen, 2019; Ipsen et al., 2021), and more importantly, not tailored for time series data (Ipsen et al., 2022). A naïve extension of Ipsen et al. (2022) for time series is likely to fail; putting the obvious difference between the static and time series data aside, the fundamental difficulty of learning the generative models for missing is that there are no explicit learning signals that could promote the model to generate "meaningful" missing values. Since we don't have ground truth for the missing values, in principle, the generative model can generate arbitrary values (e.g., zeros), and the combined classi-fier can still successfully classify time series data, which is a critical problem that is overlooked in the existing works.

To this end, we propose a hybrid model combining the deep generative models for multivariate time series data and the classification models for them. The generative model part is built under the MNAR assumption, and borrowing the structure of GRU-D (Che et al., 2018), the generative model is designed to naturally encode the continuity of the multivariate time series data in the gen-erative process. The classifier then takes the missing values generated from the generative model and executes classification, and unlike the algorithms based on heuristic imputations, it takes multi-ple feasible options for the missing values and computes predictions based on them. To tackle the difficulty in guiding the generative model to generate "meaningful" missing values, we introduce a novel regularization technique that deliberately erases *observed values* during training. As a conse-quence, the classifier is forced to do classification based more on the generated missing values, so the generative model is encouraged to produce missing values that are more advantageous for the classification. Using the various real-world multivariate time series benchmarks with missing val-ues, we demonstrate that our approach outperforms baselines both in terms of classification accuracy and uncertainty estimates.

## 2 BACKGROUND

### 2.1 SETTINGS AND NOTATIONS

Let $\boldsymbol{x} = [x_1, \ldots, x_d]^\top \in \mathbb{R}^d$ be a $d$-dimensional vector, along with the mask vector $\boldsymbol{s} = [s_1, \ldots, s_d]^\top \in \{0, 1\}^d$, where $s_j = 1$ if $x_j$ is observed and $s_j = 0$ otherwise. Given a mask $\boldsymbol{s}$, we can split $\boldsymbol{x}$ into the observed part $\boldsymbol{x}^{\mathrm{obs}} := \{x_j \mid s_j = 1\}$ and the missing part $\boldsymbol{x}^{\mathrm{mis}} := \{x_j \mid s_j = 0\}$. For a collection of data, the $i^{\mathrm{th}}$ instance is denoted as $\boldsymbol{x}_i = [x_{i,1}, \ldots, x_{i,d}]$, and $\boldsymbol{s}_i$, $\boldsymbol{x}_i^{\mathrm{obs}}$, and $\boldsymbol{x}_i^{\mathrm{mis}}$ are defined similarly. For a multivariate time-series data, we denote the vector of $t^{\mathrm{th}}$ time step as $\boldsymbol{x}_t = [x_{t,1}, \ldots, x_{t,d}] \in \mathbb{R}^d$, and the corresponding mask as $\boldsymbol{s}_t = [s_{t,1}, \ldots, s_{t,d}]$. The $t^{\mathrm{th}}$ time step of $i^{\mathrm{th}}$ instance of a collection is denoted as $\boldsymbol{x}_{t,i}$, which is split into $\boldsymbol{x}_{t,i}^{\mathrm{obs}}$ and $\boldsymbol{x}_{t,i}^{\mathrm{mis}}$ according to $\boldsymbol{s}_{t,i}$.

Following Mattei and Frellsen (2019); Ipsen et al. (2021), we assume that the joint distribution of an input $\boldsymbol{x}$ and a mask $\boldsymbol{s}$ is factorized as $p_{\boldsymbol{\theta}, \boldsymbol{\psi}}(\boldsymbol{x}, \boldsymbol{s}) = p_{\boldsymbol{\theta}}(\boldsymbol{x})p_{\boldsymbol{\psi}}(\boldsymbol{s}|\boldsymbol{x})$. The conditional distribu-tion $p_{\boldsymbol{\psi}}(\boldsymbol{s}|\boldsymbol{x})$ plays an important role for describing missing mechanism. Under MCAR assump-tion, we have $p(\boldsymbol{s}|\boldsymbol{x}) = p(\boldsymbol{s})$, under MAR we have $p_{\boldsymbol{\psi}}(\boldsymbol{s}|\boldsymbol{x}) = p_{\boldsymbol{\psi}}(\boldsymbol{s}|\boldsymbol{x}^{\mathrm{obs}})$, and under MNAR we have $p_{\boldsymbol{\psi}}(\boldsymbol{s}|\boldsymbol{x}) = p_{\boldsymbol{\psi}}(\boldsymbol{s}|\boldsymbol{x}^{\mathrm{obs}}, \boldsymbol{x}^{\mathrm{mis}})$. The likelihood for the observed data $\boldsymbol{x}^{\mathrm{obs}}$ is thus computed as $p_{\boldsymbol{\theta}, \boldsymbol{\psi}}(\boldsymbol{x}^{\mathrm{obs}}, \boldsymbol{s}) = \int p_{\boldsymbol{\theta}, \boldsymbol{\psi}}(\boldsymbol{x}, \boldsymbol{s}) \mathrm{d}\boldsymbol{x}^{\mathrm{mis}}$.

## 2.2 MISSING DATA IMPORTANCE-WEIGHTED AUTOENCODER AND ITS EXTENSIONS

In this section, we briefly review the Missing data Importance-Weighted AutoEncoder (MI-WAE) (Mattei and Frellsen, 2019), a deep generative model for missing data, and its extensions to MNAR and supervised settings. Similar to variational autoencoder (VAE) (Kingma and Welling, 2014), MIWAE assumes that a data $\boldsymbol{x}$ is genearted from a latent representation $\boldsymbol{z}$, but we only observe $\boldsymbol{x}^{\text{obs}}$ with $\boldsymbol{s}$ generated from the missing data model $p_{\boldsymbol{\psi}}(\boldsymbol{s}|\boldsymbol{x})$. MIWAE assumes MAR, so we have $p_{\boldsymbol{\psi}}(\boldsymbol{s}|\boldsymbol{x}) = p_{\boldsymbol{\psi}}(\boldsymbol{s}|\boldsymbol{x}^{\text{obs}})$. The log-likelihood for $(\boldsymbol{x}^{\text{obs}}, \boldsymbol{s})$ is then computed as

$$\log p_{\boldsymbol{\theta},\boldsymbol{\psi}}(\boldsymbol{x}^{\text{obs}}, \boldsymbol{s}) = \log \int p_{\boldsymbol{\psi}}(\boldsymbol{s}|\boldsymbol{x}^{\text{obs}})p_{\boldsymbol{\theta}}(\boldsymbol{x}^{\text{obs}}, \boldsymbol{x}^{\text{mis}}|\boldsymbol{z})p_{\boldsymbol{\theta}}(\boldsymbol{z})\mathrm{d}\boldsymbol{z}\mathrm{d}\boldsymbol{x}^{\text{mis}}$$

$$= \log p_{\boldsymbol{\psi}}(\boldsymbol{s}|\boldsymbol{x}^{\text{obs}}) + \underbrace{\log \int p_{\boldsymbol{\theta}}(\boldsymbol{x}^{\text{obs}}|\boldsymbol{z})p_{\boldsymbol{\theta}}(\boldsymbol{z})\mathrm{d}\boldsymbol{z}}_{=\log p_{\boldsymbol{\theta}}(\boldsymbol{x}^{\text{obs}})} . \tag{1}$$

For the missing data imputation, $p_{\boldsymbol{\psi}}(\boldsymbol{s}|\boldsymbol{x}^{\text{obs}})$ is not necessary, so we choose to maximize only the $\log p_{\boldsymbol{\theta}}(\boldsymbol{x}^{\text{obs}})$. The integral is intractable, so we consider the Importance Weighted AutoEncoder (IWAE) lower bound (Burda et al., 2015),

$$\log p_{\boldsymbol{\theta}}(\boldsymbol{x}^{\text{obs}}) \geq \mathbb{E}_{\boldsymbol{z}_{1:K}}\left[\log \frac{1}{K}\sum_{k=1}^{K}\frac{p_{\boldsymbol{\theta}}(\boldsymbol{x}^{\text{obs}}|\boldsymbol{z}_k)p_{\boldsymbol{\theta}}(\boldsymbol{z}_k)}{q_{\boldsymbol{\phi}}(\boldsymbol{z}_k|\boldsymbol{x}^{\text{obs}})}\right] := \mathcal{L}_{\text{MIWAE}}(\boldsymbol{\theta}, \boldsymbol{\phi}). \tag{2}$$

Here, $q_{\boldsymbol{\phi}}(\boldsymbol{z}_k|\boldsymbol{x}^{\text{obs}})$ for $k = 1, \ldots, K$ are i.i.d. copies of the variational distribution (encoder) $q_{\boldsymbol{\phi}}(\boldsymbol{z}|\boldsymbol{x}^{\text{obs}})$ approximating the true posterior $p_{\boldsymbol{\theta}}(\boldsymbol{z}|\boldsymbol{x}^{\text{obs}})$. $\mathbb{E}_{\boldsymbol{z}_{1:K}}$ denotes the expectation w.r.t. $\prod_{k=1}^{K} q_{\boldsymbol{\phi}}(\boldsymbol{z}_k|\boldsymbol{x}^{\text{obs}})$. $K$ is the number of particles, and the lower bound gets tighter as $K$ increases and converges to the upper-bound as $K \to \infty$.

Ipsen et al. (2021) presented not-MIWAE, an extension of MIWAE with MNAR assumption. The log-likelihood for $(\boldsymbol{x}^{\text{obs}}, \boldsymbol{s})$ under the MNAR assumption is,

$$\log p_{\boldsymbol{\theta},\boldsymbol{\psi}}(\boldsymbol{x}^{\text{obs}}, \boldsymbol{s}) = \log \int p_{\boldsymbol{\psi}}(\boldsymbol{s}|\boldsymbol{x}^{\text{obs}}, \boldsymbol{x}^{\text{mis}})p_{\boldsymbol{\theta}}(\boldsymbol{x}^{\text{obs}}|\boldsymbol{z})p_{\boldsymbol{\theta}}(\boldsymbol{x}^{\text{mis}}|\boldsymbol{z})p_{\boldsymbol{\theta}}(\boldsymbol{z})\mathrm{d}\boldsymbol{z}\mathrm{d}\boldsymbol{x}^{\text{mis}}, \tag{3}$$

where we are assuming that $(\boldsymbol{x}^{\text{obs}}, \boldsymbol{x}^{\text{mis}})$ are independent given $\boldsymbol{z}$. The corresponding IWAE lower-bound with the variational distribution $q_{\boldsymbol{\phi}}(\boldsymbol{x}^{\text{mis}}, \boldsymbol{z}|\boldsymbol{x}^{\text{obs}}) = p_{\boldsymbol{\theta}}(\boldsymbol{x}^{\text{mis}}|\boldsymbol{z})q_{\boldsymbol{\phi}}(\boldsymbol{z}|\boldsymbol{x}^{\text{obs}})$ is,

$$\mathcal{L}_{\text{notMIWAE}}(\boldsymbol{\theta}, \boldsymbol{\psi}, \boldsymbol{\phi}) := \mathbb{E}_{\boldsymbol{z}_{1:K}, \boldsymbol{x}^{\text{mis}}_{1:K}}\left[\log \frac{1}{K}\sum_{k=1}^{K}\frac{p_{\boldsymbol{\theta}}(\boldsymbol{s}|\boldsymbol{x}^{\text{obs}}, \boldsymbol{x}^{\text{mis}}_k)p_{\boldsymbol{\theta}}(\boldsymbol{x}^{\text{obs}}|\boldsymbol{z}_k)p_{\boldsymbol{\theta}}(\boldsymbol{z}_k)}{q_{\boldsymbol{\phi}}(\boldsymbol{z}_k|\boldsymbol{x}^{\text{obs}})} . \right], \tag{4}$$

where $\mathbb{E}_{\boldsymbol{z}_{1:K}, \boldsymbol{x}^{\text{mis}}_{1:K}}$ denotes the expectation w.r.t. $\prod_{k=1}^{K} p_{\boldsymbol{\theta}}(\boldsymbol{x}^{\text{mis}}_k|\boldsymbol{z}_k)q_{\boldsymbol{\phi}}(\boldsymbol{z}_k|\boldsymbol{x}^{\text{obs}})$.

On the other hand, Ipsen et al. (2022) extended MIWAE to a supervised learning setting, where the goal is to learn the joint distribution of an observed input $\boldsymbol{x}^{\text{obs}}$, a mask $\boldsymbol{s}$, and corresponding label $\boldsymbol{y}$,

$$\log p_{\boldsymbol{\theta},\boldsymbol{\psi},\boldsymbol{\lambda}}(\boldsymbol{y}, \boldsymbol{x}^{\text{obs}}, \boldsymbol{s})$$

$$= \log \int p_{\boldsymbol{\lambda}}(\boldsymbol{y}|\boldsymbol{x}^{\text{obs}}, \boldsymbol{x}^{\text{mis}})p_{\boldsymbol{\psi}}(\boldsymbol{s}|\boldsymbol{x}^{\text{obs}}, \boldsymbol{x}^{\text{mis}})p_{\boldsymbol{\theta}}(\boldsymbol{x}^{\text{obs}}, \boldsymbol{x}^{\text{mis}}|\boldsymbol{z})p_{\boldsymbol{\theta}}(\boldsymbol{z})\mathrm{d}\boldsymbol{z}\mathrm{d}\boldsymbol{x}^{\text{mis}}$$

$$= \log p_{\boldsymbol{\psi}}(\boldsymbol{s}|\boldsymbol{x}^{\text{obs}}) + \underbrace{\log \int p_{\boldsymbol{\lambda}}(\boldsymbol{y}|\boldsymbol{x}^{\text{obs}}, \boldsymbol{x}^{\text{mis}})p_{\boldsymbol{\theta}}(\boldsymbol{x}^{\text{obs}}|\boldsymbol{z})p_{\boldsymbol{\theta}}(\boldsymbol{x}^{\text{mis}}|\boldsymbol{z})p_{\boldsymbol{\theta}}(\boldsymbol{z})\mathrm{d}\boldsymbol{z}}_{=\log p_{\boldsymbol{\theta},\boldsymbol{\lambda}}(\boldsymbol{y}, \boldsymbol{x}^{\text{obs}})}, \tag{5}$$

The term $p_{\boldsymbol{\psi}}(\boldsymbol{s}|\boldsymbol{x}^{\text{obs}})$ is irrelevant to the prediction for $\boldsymbol{y}$, so we choose to maximize $\log p_{\boldsymbol{\theta},\boldsymbol{\lambda}}(\boldsymbol{y}, \boldsymbol{x}^{\text{obs}})$, which again can be lower-bounded by IWAE bound with the variational distribution $q_{\boldsymbol{\phi}}(\boldsymbol{z}, \boldsymbol{x}^{\text{mis}}|\boldsymbol{x}^{\text{obs}}) = p_{\boldsymbol{\theta}}(\boldsymbol{x}^{\text{mis}}|\boldsymbol{z})q_{\boldsymbol{\phi}}(\boldsymbol{z}|\boldsymbol{x}^{\text{obs}})$:

$$\mathcal{L}_{\text{supMIWAE}}(\boldsymbol{\theta}, \boldsymbol{\lambda}, \boldsymbol{\phi}) := \mathbb{E}_{\boldsymbol{z}_{1:K}, \boldsymbol{x}^{\text{mis}}_{1:K}}\left[\log \frac{1}{K}\sum_{k=1}^{K}\frac{p_{\boldsymbol{\lambda}}(\boldsymbol{y}|\boldsymbol{x}^{\text{obs}}, \boldsymbol{x}^{\text{mis}}_k)p_{\boldsymbol{\theta}}(\boldsymbol{x}^{\text{obs}}|\boldsymbol{z}_k)p(\boldsymbol{z}_k)}{q_{\boldsymbol{\phi}}(\boldsymbol{z}_k|\boldsymbol{x}^{\text{obs}})}\right], \tag{6}$$

where $\mathbb{E}_{\boldsymbol{z}_{1:K}, \boldsymbol{x}^{\text{mis}}_{1:K}}$ denotes the expectation w.r.t. $\prod_{k=1}^{K} p_{\boldsymbol{\theta}}(\boldsymbol{x}^{\text{mis}}_k|\boldsymbol{z}_k)q_{\boldsymbol{\phi}}(\boldsymbol{z}_k|\boldsymbol{x}^{\text{obs}})$.

## 2.3 GRU FOR MULTIVARIATE TIME SERIES DATA AND IMPUTATION METHODS

We briefly review GRU (Cho et al., 2014) as it is being used as a building block for our method. Given a multivariate time series $(\boldsymbol{x}_t)_{t=1}^T$, GRU takes a vector of one time step at a time and accumulates the information into a hidden state $\boldsymbol{h}_t$. Specifically, the forward pass at $t^{\text{th}}$ time step takes $\boldsymbol{x}_t$ and updates the hidden state $\boldsymbol{h}_t$ as follows:

$$\boldsymbol{a}_t = \sigma(\boldsymbol{W_a}\boldsymbol{x}_t + \boldsymbol{U_a}\boldsymbol{h}_{t-1} + \boldsymbol{b_a}), \quad \boldsymbol{r}_t = \sigma(\boldsymbol{W_r}\boldsymbol{x}_t + \boldsymbol{U_r}\boldsymbol{h}_{t-1} + \boldsymbol{b_r})$$

$$\tilde{\boldsymbol{h}}_t = \tanh(\boldsymbol{W}\boldsymbol{x}_t + \boldsymbol{U}(\boldsymbol{r}_t \odot \boldsymbol{h}_{t-1}) + \boldsymbol{b}), \quad \boldsymbol{h}_t = (1 - \boldsymbol{a}_t) \odot \boldsymbol{h}_{t-1} + \boldsymbol{a}_t \odot \tilde{\boldsymbol{h}}_t,$$

where $\odot$ denotes the element-wise multiplication. We also review the heuristical imputation methods described in Che et al. (2018), which are for GRU-based multivariate time-series classifiers and commonly used as baselines.

- **GRU-zero**: simply put zero for all missing values. That is, $\hat{x}_{t,j} = s_{t,j}x_{t,j}$.

- **GRU-mean**: imputes the missing values as $\hat{x}_{t,j} = s_{t,j}x_{t,j} + (1 - s_{t,j})\bar{x}_j$, where $\bar{x}_j = \sum_{i=1}^n \sum_{t=1}^T s_{t,i,j}x_{t,i,j} / \sum_{i=1}^n \sum_{t=1}^T s_{t,i,j}$ is the empirical mean of observed values for $j^{\text{th}}$ feature of a given collection of time series data $((\boldsymbol{x}_{t,i})_{t=1}^T)_{i=1}^n$.

- **GRU-forward**: imputes the missing values as $\hat{x}_{t,j} = s_{t,j}x_{t,j} + (1 - s_{t,j})x_{t',j}$, where $t'$ is the last time when $j^{\text{th}}$ feature was observed before $t$.

- **GRU-simple**: along with the imputed vector $\hat{\boldsymbol{x}}_t$ (either by GRU-mean or GRU-forward), concatenate additional information. Che et al. (2018) proposed to concatenate 1) the mask $\boldsymbol{s}_t$, and the *time-interval* $\boldsymbol{\delta}_t$ saving the length of the intervals between observed values (see Che et al. (2018) for precise definition). The concatenated vector $[\hat{\boldsymbol{x}}_t, \boldsymbol{s}_t, \boldsymbol{\delta}_t]$ is then fed into GRU.

- **GRU-D**: introduces *learnable decay* values for the input $\boldsymbol{x}_t$ and hidden state $\boldsymbol{h}_t$ as follows:

$$\boldsymbol{\gamma_x} = \exp(-\max(\boldsymbol{W_{\gamma_x}}\boldsymbol{\delta}_t + \boldsymbol{b_{\gamma_x}}, \boldsymbol{0})), \quad \boldsymbol{\gamma_h} = \exp(-\max(\boldsymbol{W_{\gamma_h}}\boldsymbol{\delta}_t + \boldsymbol{b_{\gamma_h}}, \boldsymbol{0})). \quad (7)$$

Given a vector $\boldsymbol{x}_t$ with mask $\boldsymbol{s}_t$, GRU-D imputes the missing values as

$$\hat{x}_{t,j} = s_{t,j}x_{t,j} + (1 - s_{t,j})(\gamma_{\boldsymbol{x},t}x_{t',j} + (1 - \gamma_{\boldsymbol{x},t})\bar{x}_j). \quad (8)$$

That is, the missing is imputed as a mixture of the last observed $x_{t',j}$ and the empirical mean $\bar{x}_j$ with the mixing coefficient set as the learned decay. The hidden state from the previous time step $\boldsymbol{h}_{t-1}$ is decayed as $\boldsymbol{\gamma_h} \odot \boldsymbol{h}_{t-1}$ and passed through GRU with the imputed $\hat{\boldsymbol{x}}_t$.

## 3 METHODS

In this section, we describe our method, a probabilistic framework for multivariate time series data with missing values. Our method is an extension of supMIWAE to time series data under MNAR assumption, but the actual implementation is not merely a naïve composition of the existing models. In Section 3.1, we first present supnotMIWAE, an MNAR version of supMIWAE, with the encoder and decoder architectures designed for time series data with missing values. In Section 3.2, we show why the sup(not)MIWAE for data with missings may fail, and propose a novel regularization technique to prevent that.

### 3.1 SUPNOTMIWAE FOR MULTIVARIATE TIME SERIES DATA

Given a multivariate time series data $\boldsymbol{x}_{1:T} := (\boldsymbol{x}_t)_{t=1}^T$ with observed $\boldsymbol{x}_{1:T}^{\text{obs}}$ and missing $\boldsymbol{x}_{1:T}^{\text{mis}}$, a missing mask $\boldsymbol{s}_{1:T} := (\boldsymbol{s}_t)_{t=1}^T$ and a label $\boldsymbol{y}$, we assume the following state-space model with latent vectors $\boldsymbol{z}_{1:T} := (\boldsymbol{z}_t)_{t=1}^T$.

$$p_{\boldsymbol{\theta},\boldsymbol{\psi},\boldsymbol{\lambda}}(\boldsymbol{y}, \boldsymbol{x}_{1:T}^{\text{obs}}, \boldsymbol{s}_{1:T})$$

$$= \int p_{\boldsymbol{\lambda}}(\boldsymbol{y}|\boldsymbol{x}_{1:T}^{\text{obs}}, \boldsymbol{x}_{1:T}^{\text{mis}})p_{\boldsymbol{\theta}}(\boldsymbol{x}_{1:T}^{\text{obs}}|\boldsymbol{z}_{1:T})p_{\boldsymbol{\theta}}(\boldsymbol{x}_{1:T}^{\text{mis}}|\boldsymbol{z}_{1:T})p_{\boldsymbol{\theta}}(\boldsymbol{z}_{1:T})p_{\boldsymbol{\psi}}(\boldsymbol{s}_{1:T}|\boldsymbol{x}_{1:T})\mathrm{d}\boldsymbol{x}_{1:T}^{\text{mis}}\mathrm{d}\boldsymbol{z}_{1:T}. \quad (9)$$

Below we describe each component more in detail.

**Prior** $p_{\boldsymbol{\theta}}(\boldsymbol{z}_{1:T})$    we assume an autoregressive prior for $\boldsymbol{z}_{1:T}$,

$$p_{\boldsymbol{\theta}}(\boldsymbol{z}_{1:T}) = \mathcal{N}(\boldsymbol{z}_1|\boldsymbol{0}, \boldsymbol{I}) \prod_{t=2}^{T} \mathcal{N}(\boldsymbol{z}_t|\boldsymbol{\mu}_{\mathrm{pr}}(\boldsymbol{z}_{1:t-1}), \mathrm{diag}(\boldsymbol{\sigma}_{\mathrm{pr}}^2(\boldsymbol{z}_{1:t-1})), \tag{10}$$

where $(\boldsymbol{\mu}_{\mathrm{pr}}(\boldsymbol{z}_{1:t}), \sigma_{\mathrm{pr}}(\boldsymbol{z}_{1:t}))_{t=1}^{T-1}$ are computed as

$$\boldsymbol{h}_t = \mathrm{GRU}_{\mathrm{pr}}(\boldsymbol{z}_t, \boldsymbol{h}_{t-1}), \quad \boldsymbol{\mu}_{\mathrm{pr}}(\boldsymbol{z}_{1:t}), \boldsymbol{\sigma}_{\mathrm{pr}}(\boldsymbol{z}_{1:t}) = \mathrm{MLP}_{\mathrm{pr}}(\boldsymbol{h}_{t-1}). \tag{11}$$

Here, $\mathrm{GRU}_{\mathrm{pr}}(\boldsymbol{z}_t, \boldsymbol{h}_{t-1})$ is a GRU cell that takes $\boldsymbol{z}_t$ and the hidden state $\boldsymbol{h}_{t-1}$ and update it to $\boldsymbol{h}_t$.

**Decoders** $p_{\boldsymbol{\theta}}(\boldsymbol{x}_{1:T}^{\mathbf{obs}}|\boldsymbol{z}_{1:T})$ **and** $p_{\boldsymbol{\theta}}(\boldsymbol{x}_{1:T}^{\mathbf{mis}}|\boldsymbol{z}_{1:T})$    The decoder for the observed $p_{\boldsymbol{\theta}}(\boldsymbol{x}_{1:T}^{\mathrm{obs}}|\boldsymbol{z}_{1:T})$ is defined in autoregressive fashion,

$$p_{\boldsymbol{\theta}}(\boldsymbol{x}_{1:T}^{\mathrm{obs}}|\boldsymbol{z}_{1:T}) = \prod_{t=1}^{T} \mathcal{N}(\boldsymbol{x}_t^{\mathrm{obs}}|\boldsymbol{\mu}_{\mathrm{dec}}(\boldsymbol{z}_{1:t}), \mathrm{diag}(\boldsymbol{\sigma}_{\mathrm{dec}}^2(\boldsymbol{z}_{1:t}))), \tag{12}$$

where $(\boldsymbol{\mu}_{\mathrm{dec}}(\boldsymbol{z}_{1:t}), \boldsymbol{\sigma}_{\mathrm{dec}}(\boldsymbol{z}_{1:t}))_{t=1}^{T}$ are defined as in (11) using GRU. The decoder for the missing $p_{\boldsymbol{\theta}}(\boldsymbol{x}_{1:T}^{\mathrm{mis}}|\boldsymbol{z}_{1:T})$ shares the same model; that is, both observed value decoder $p_{\boldsymbol{\theta}}(\boldsymbol{x}_{1:T}^{\mathrm{obs}}|\boldsymbol{z}_{1:T})$ and the missing decoder $p_{\boldsymbol{\theta}}(\boldsymbol{x}_{1:T}^{\mathrm{mis}}|\boldsymbol{z}_{1:T})$ share the same model.

**Missing model** $p_{\boldsymbol{\psi}}(\boldsymbol{s}_{1:T}|\boldsymbol{x}_{1:T})$    The missing model is simply assumed to be independent Bernoulli distributions over the time steps and features.

$$p_{\boldsymbol{\psi}}(\boldsymbol{s}_{1:T}|\boldsymbol{x}_{1:T}) = \prod_{t=1}^{T} \prod_{j=1}^{d} \mathrm{Bern}(s_{t,j}|\sigma_{\mathrm{mis},t,j}(\boldsymbol{x}_{1:T})), \tag{13}$$

where $\sigma_{\mathrm{mis}}(\boldsymbol{x}_{1:T})$ is computed as

$$\sigma_{\mathrm{mis}}(\boldsymbol{x}_{1:T}) = \mathrm{MLP}_{\mathrm{mis}}(\boldsymbol{x}_{1:T}). \tag{14}$$

**Classifier** $p_{\boldsymbol{\lambda}}(\boldsymbol{y}|\boldsymbol{x}_{1:T}^{\mathbf{obs}}, \boldsymbol{x}_{1:T}^{\mathbf{mis}})$    We simply use a common GRU-based time series classifier for this. Let $\boldsymbol{h}_T$ be the hidden state from a GRU after consuming $(\boldsymbol{x}_{1:T}^{\mathrm{obs}}, \boldsymbol{x}_{1:T}^{\mathrm{mis}})$. Then the conditional distribution is defined as

$$p_{\boldsymbol{\lambda}}(\boldsymbol{y}|\boldsymbol{x}_{1:T}^{\mathrm{obs}}, \boldsymbol{x}_{1:T}^{\mathrm{mis}}) = \mathrm{Categorical}(\boldsymbol{y}\,|\,\mathrm{Softmax}(\mathrm{Linear}_{\mathrm{cls}}(\boldsymbol{h}_T)). \tag{15}$$

During the forward pass, the classifier takes the observed input $\boldsymbol{x}_{1:T}^{\mathrm{obs}}$ and the missing values *generated* from the decoder $p_{\boldsymbol{\theta}}(\boldsymbol{x}_{1:T}^{\mathrm{mis}}|\boldsymbol{z}_{1:T})$. We find it beneficial to adopt the idea of GRU-D, where instead of directly putting the generated missing values $\boldsymbol{x}_{1:T}^{\mathrm{mis}}$, putting the *decayed* missing values as follows:

$$\tilde{\boldsymbol{x}}_t := (\boldsymbol{x}_t^{\mathrm{obs}}, \boldsymbol{x}_t^{\mathrm{mis}}) \text{ where } \boldsymbol{x}_t^{\mathrm{mis}} \sim p_{\boldsymbol{\theta}}(\boldsymbol{x}_t^{\mathrm{mis}}|\boldsymbol{z}_{1:T}), \tag{16}$$

$$\hat{x}_{t,j} = s_{t,j} x_{t,j} + (1 - s_{t,j})(\gamma_{\mathrm{cls},t} x_{t',j} + (1 - \gamma_{\mathrm{cls},t})\tilde{x}_{t,j}), \tag{17}$$

where $\boldsymbol{\gamma}_{\mathrm{cls}} = \exp(-\max(\boldsymbol{0}, \boldsymbol{W}_{\mathrm{cls}}\boldsymbol{\delta}_t + \boldsymbol{b}_{\mathrm{cls}}))$ is a learnable decay. We find this stabilizes the learning when the generated missing values $\boldsymbol{x}_{1:T}^{\mathrm{mis}}$ are inaccurate, for instance, in the early stage of learning. Note also the difference between (16) and the original GRU-D imputation (8). In GRU-D, the last observed values are mixed with the mean feature, while ours mix them with the generated values.

**Encoder** $q_{\boldsymbol{\phi}}(\boldsymbol{z}_{1:T}|\boldsymbol{x}_{1:T}^{\mathbf{obs}})$    Given the generative model defined as above, we introduce the variational distribution for $(\boldsymbol{x}_{1:T}^{\mathrm{mis}}, \boldsymbol{z}_{1:T})$ for lower-bounding the log-likelihood.

$$q_{\boldsymbol{\theta},\boldsymbol{\phi}}(\boldsymbol{x}_{1:T}^{\mathrm{mis}}, \boldsymbol{z}_{1:T}|\boldsymbol{x}_{1:T}^{\mathrm{obs}}) = p_{\boldsymbol{\theta}}(\boldsymbol{x}_{1:T}^{\mathrm{mis}}|\boldsymbol{z}_{1:T})q_{\boldsymbol{\phi}}(\boldsymbol{z}_{1:T}|\boldsymbol{x}_{1:T}^{\mathrm{obs}}). \tag{18}$$

Here, the encoder $q_{\boldsymbol{\phi}}(\boldsymbol{z}_{1:T}|\boldsymbol{x}_{1:T}^{\mathrm{obs}})$ is defined as an autoregressive model as before,

$$q_{\boldsymbol{\phi}}(\boldsymbol{z}_{1:T}|\boldsymbol{x}_{1:T}^{\mathrm{obs}}) = \prod_{t=1}^{T} \mathcal{N}(\boldsymbol{z}_t|\boldsymbol{\mu}_{\mathrm{enc}}(\boldsymbol{x}_{1:t}^{\mathrm{obs}}), \mathrm{diag}(\boldsymbol{\sigma}_{\mathrm{enc}}^2(\boldsymbol{x}_{1:t}^{\mathrm{obs}}))). \tag{19}$$

Similar to the decoder, we use GRU to compute $(\boldsymbol{\mu}_{\mathrm{enc}}(\boldsymbol{x}_{1:t}^{\mathrm{obs}}), \boldsymbol{\sigma}_{\mathrm{enc}}(\boldsymbol{x}_{1:t}^{\mathrm{obs}}))$ for $t = 1, \ldots, T$. However, since $\boldsymbol{x}_t$ includes many missing values, rather than putting only the observed values, we find it beneficial to put the imputed value $\hat{\boldsymbol{x}}_t$ as an input to the encoder. For the imputation, we adopt GRU-D. To summarize, the encoder parameters are constructed from GRU-D outputs, with the inputs imputed with learnable decay values.

**Objective** Having all the ingredients defined, the IWAE bound for supnotMIWAE is computed as follows:

$$\log p_{\boldsymbol{\lambda},\boldsymbol{\theta},\boldsymbol{\psi}}(\boldsymbol{y}, \boldsymbol{x}_{1:T}^{\text{obs}}, \boldsymbol{s}_{1:T}) \geq \mathcal{L}_{\text{supnotMIWAE}}(\boldsymbol{\lambda},\boldsymbol{\theta},\boldsymbol{\psi},\boldsymbol{\phi}) := \mathbb{E}_{\boldsymbol{z}_{1:K,1:T},\boldsymbol{x}_{1:K,1:T}^{\text{mis}}}\left[\log \frac{1}{K}\sum_{k=1}^{K}\omega_k\right], \quad (20)$$

where

$$\omega_k := \frac{p_{\boldsymbol{\theta}}(\boldsymbol{y}|\boldsymbol{x}_{1:T}^{\text{obs}}, \boldsymbol{x}_{k,1:T}^{\text{mis}})p_{\boldsymbol{\psi}}(\boldsymbol{s}_{1:T}|\boldsymbol{x}_{1:T}^{\text{obs}}, \boldsymbol{x}_{k,1:T}^{\text{mis}})p_{\boldsymbol{\theta}}(\boldsymbol{x}_{1:T}^{\text{obs}}|\boldsymbol{z}_{k,1:T})p_{\boldsymbol{\theta}}(\boldsymbol{z}_{k,1:T})}{q_{\boldsymbol{\phi}}(\boldsymbol{z}_{k,1:T}|\boldsymbol{x}_{1:T}^{\text{obs}})}. \quad (21)$$

Here, $(q_{\boldsymbol{\phi}}(\boldsymbol{z}_{k,1:T}|\boldsymbol{x}_{1:T}^{\text{obs}})p_{\boldsymbol{\theta}}(\boldsymbol{x}_{k,1:T}^{\text{mis}}|\boldsymbol{z}_{k,1:T}))_{k=1}^{K}$ are i.i.d. copies of the variational distribution, and $\mathbb{E}_{\boldsymbol{z}_{1:K,1:T},\boldsymbol{x}_{1:K,1:T}^{\text{mis}}}$ denotes the expectation w.r.t. those i.i.d. copies.

## 3.2 OBSDROPOUT: REGULARIZING SUPNOTMIWAE FOR BETTER IMPUTATION

The problem with (20) is that there is no clear supervison for the missing values $\boldsymbol{x}_{1:T}^{\text{mis}}$. Obviously, if we had an access to the missing values, the conditional probability $p_{\boldsymbol{\theta}}(\boldsymbol{x}_{1:T}^{\text{mis}}|\boldsymbol{z}_{1:T})$ would guide the model to learn to correctly impute those missing values. Without such true values, we can only encourage the model to impute the missing values with some indirect criteria. In the objective (20), there are two terms that the model hinges on for this matter.

- The missing model $p_{\boldsymbol{\psi}}(\boldsymbol{s}_{1:T}|\boldsymbol{x}_{1:T}^{\text{obs}}, \boldsymbol{x}_{1:T}^{\text{mis}})$: this term encourages the model to reconstruct the missing mask $s_t$ from the imputed value $x_t^{\text{mis}}$, so in principle, the model should impute the missing values in a way that they are distinguishable from the observed values. However, in general, the distributions of the observed and the missings are not necessarily different, and more importantly, the model can easily cheat the objective. For instance, consider a trivial case where the model imputes all the missing values with zero. The conditional probability $p_{\boldsymbol{\psi}}(\boldsymbol{s}_{1:T}|\boldsymbol{x}_{1:T}^{\text{obs}}, \boldsymbol{x}_{1:T}^{\text{mis}})$ can still be maximized by setting $\sigma_{\text{mis}}(x_{t,j}) = 0$ if $x_{t,j} = 0$ (unless there are not many observed with $x_{t,j}^{\text{obs}} = 0$).

- The classifier $p_{\boldsymbol{\theta}}(\boldsymbol{y}|\boldsymbol{x}_{1:T}^{\text{obs}}, \boldsymbol{x}_{1:T}^{\text{mis}})$: this term expects the model to generate meaningful imputations so that they are helpful for the classification. However, as shown in prior works (Che et al., 2018), the classifier can achieve decent classification accuracy *without* meaningful imputations, for instance, it will still be able to classify the signals while all the missing values are imputed with zeros. Hence, in the current form, there is no strong incentive for the model to learn non-trivial imputations that will bring significant accuracy gain over the zero imputations.

To summarize, in the current form, the objective (20) is not likely to generate realistic missing values. To resolve this, we may introduce a missing model $p_{\boldsymbol{\theta}}(\boldsymbol{s}_{1:T}|\boldsymbol{x}_{1:T}^{\text{obs}}, \boldsymbol{x}_{1:T}^{\text{mis}})$ much more elaborated than the simple i.i.d. model that we are using right now, but that may require some dataset-specific design. Instead, we present a simple regularization technique that can effectively enhance the quality of the imputed values.

Our idea is simple; when passing the observed inputs $\boldsymbol{x}_{1:T}^{\text{obs}}$ and the imputed missing values $\hat{\boldsymbol{x}}_{1:T}^{\text{mis}}$ (i.e., imputed by (16)) to the classifier, *deliberately drop* some portion of the observed inputs. Without dropping the observed inputs, the classifier may heavily rely on the observed inputs to do the classification, but if some of the observed inputs are dropped out during training, the classifier can focus more on the imputed missing values $\hat{\boldsymbol{x}}_{1:T}^{\text{mis}}$. As a result, the model is encouraged to generate more "useful" missing values that are beneficial for classification. More specifically, let $\beta$ be a predefined dropout probability. Then we construct the imputed input $\hat{\boldsymbol{x}}_t$ to the classifier as follows:

$$m_{t,j} \sim \text{Bern}(1-\beta), \quad \tilde{s}_{t,j} := s_{t,j}m_{t,j}$$
$$\tilde{\tilde{\boldsymbol{x}}}_t := (\boldsymbol{x}_{1:T}^{\text{obs}}, \boldsymbol{x}_{1:T}^{\text{mis}}) \text{ where } (\boldsymbol{x}_{1:T}^{\text{obs}}, \boldsymbol{x}_{1:T}^{\text{mis}}) \sim p_{\boldsymbol{\theta}}(\boldsymbol{x}_{1:T}^{\text{obs}}, \boldsymbol{x}_{1:T}^{\text{mis}}|\boldsymbol{z}_{1:T}),$$
$$\hat{x}_{t,j} := \tilde{s}_{t,j}x_{t,j} + (1-\tilde{s}_{t,j})(\gamma_{\text{cls},t}x_{t',j} + (1-\gamma_{\text{cls},t})\tilde{\tilde{x}}_{t,j}). \quad (22)$$

That is, when an observed $x_{t,j}$ is dropped out, we put a generated value with the decay applied as in (16), so that the classifier could focus more on the values generated by the decoder as we intended. We call this idea *ObsDropout*, since we are dropping out the observed values during the training.

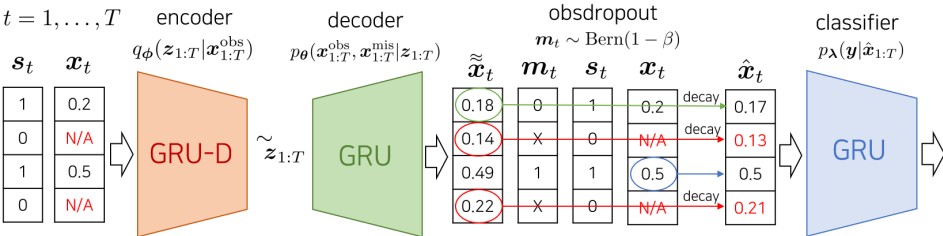

**Figure 1:** An overview of our model with obsdropout.

With the mask variables $\boldsymbol{m}_{1:T}$, the likelihood is obtained by marginalizing $\boldsymbol{m}_{1:T}$ from the joint distribution with $\boldsymbol{m}_{1:T}$.

$$p_{\boldsymbol{\theta},\boldsymbol{\psi},\boldsymbol{\lambda}}(\boldsymbol{y},\boldsymbol{x}_{1:T}^{\text{obs}},\boldsymbol{s}_{1:T}) = \int p_{\boldsymbol{\theta},\boldsymbol{\psi},\boldsymbol{\lambda}}(\boldsymbol{y},\boldsymbol{x}_{1:T}^{\text{obs}},\boldsymbol{s}_{1:T},\boldsymbol{m}_{1:T})\mathrm{d}\boldsymbol{m}_{1:T}, \tag{23}$$

where the joint distribution is decomposed as

$$p_{\boldsymbol{\theta},\boldsymbol{\psi},\boldsymbol{\lambda}}(\boldsymbol{y},\boldsymbol{x}_{1:T}^{\text{obs}},\boldsymbol{s}_{1:T},\boldsymbol{m}_{1:T})$$
$$= p_{\boldsymbol{\lambda}}(\boldsymbol{y}|\boldsymbol{x}_{1:T}^{\text{obs}},\boldsymbol{x}_{1:T}^{\text{mis}},\boldsymbol{m}_{1:T})p_{\beta}(\boldsymbol{m}_{1:T})p_{\boldsymbol{\psi}}(\boldsymbol{s}_{1:T}|\boldsymbol{x}_{1:T})p_{\boldsymbol{\theta}}(\boldsymbol{x}_{1:T}^{\text{obs}}|\boldsymbol{z}_{1:T})p_{\boldsymbol{\theta}}(\boldsymbol{x}_{1:T}^{\text{mis}}|\boldsymbol{z}_{1:T})p_{\boldsymbol{\theta}}(\boldsymbol{z}_{1:T}). \tag{24}$$

Consequently, the IWAE objective is slightly modified as follows:

$$\mathcal{L}'_{\text{supnotMIWAE}}(\boldsymbol{\lambda},\boldsymbol{\theta},\boldsymbol{\psi},\boldsymbol{\phi}) := \mathbb{E}_{\boldsymbol{z}_{1:K,1:T},\boldsymbol{x}_{1:K,1:T}^{\text{mis}},\boldsymbol{m}_{1:K}}\left[\log\frac{1}{K}\sum_{k=1}^{K}\omega_k\right], \tag{25}$$

where

$$\omega_k := \frac{p_{\boldsymbol{\lambda}}(\boldsymbol{y}|\boldsymbol{x}_{1:T}^{\text{obs}},\boldsymbol{x}_{k,1:T}^{\text{mis}},\boldsymbol{m}_{k,1:T})p_{\boldsymbol{\psi}}(\boldsymbol{s}_{1:T}|\boldsymbol{x}_{1:T}^{\text{obs}},\boldsymbol{x}_{k,1:T}^{\text{mis}})p_{\boldsymbol{\theta}}(\boldsymbol{x}_{1:T}^{\text{obs}}|\boldsymbol{z}_{k,1:T})p_{\boldsymbol{\theta}}(\boldsymbol{z}_{k,1:T})}{q_{\boldsymbol{\phi}}(\boldsymbol{z}_{k,1:T}|\boldsymbol{x}_{1:T}^{\text{obs}})}, \tag{26}$$

where the expectation is over $K$ i.i.d. copies of the variational distribution,

$$q(\boldsymbol{z}_{1:T},\boldsymbol{x}_{1:T}^{\text{mis}},\boldsymbol{m}_{1:T}|\boldsymbol{x}_{1:T}^{\text{obs}}) = q_{\boldsymbol{\phi}}(\boldsymbol{z}_{1:T}|\boldsymbol{x}_{1:T}^{\text{obs}})p_{\boldsymbol{\theta}}(\boldsymbol{x}_{1:T}^{\text{mis}}|\boldsymbol{z}_{1:T})p_{\beta}(\boldsymbol{m}_{1:T}), \tag{27}$$

with $p_{\beta}(\boldsymbol{m}_{1:T}) := \prod_{t=1}^{T}\prod_{j=1}^{d}\text{Bern}(m_{t,j}|\beta)$.

### 3.3 PREDICTION

Similar to SupMIWAE, we exploit Self-Normalized Importance Sampling (SNIS) to approximate the predictive distribution for a new input $\boldsymbol{x}_{1:T}^{\text{obs}}$. With the model trained with obsdropout, we have

$$p(\boldsymbol{y}|\boldsymbol{x}_{1:T}^{\text{obs}}) \approx \frac{1}{S}\sum_{s=1}^{S}\sum_{k=1}^{K}\bar{\omega}_k^{(s)}p_{\boldsymbol{\lambda}}(\boldsymbol{y}|\boldsymbol{x}_{1:T}^{\text{obs}},(\boldsymbol{x}^{\text{mis}})_{k,1:T}^{(s)},\boldsymbol{m}_{k,1:T}^{(s)}), \tag{28}$$

where

$$(\boldsymbol{z}_{k,1:T}^{(s)},(\boldsymbol{x}^{\text{mis}})_{k,1:T}^{(s)},\boldsymbol{m}_{k,1:T}^{(s)}) \overset{\text{i.i.d.}}{\sim} q_{\boldsymbol{\phi}}(\boldsymbol{z}_{1:T}|\boldsymbol{x}_{1:T}^{\text{obs}})p_{\boldsymbol{\theta}}(\boldsymbol{x}_{1:T}^{\text{mis}}|\boldsymbol{z}_{1:T})p_{\beta}(\boldsymbol{m}_{1:T}),$$
$$\omega_k := \frac{p_{\boldsymbol{\theta}}(\boldsymbol{x}_{1:T}^{\text{obs}}|\boldsymbol{z}_{k,1:T})p_{\boldsymbol{\theta}}(\boldsymbol{z}_{k,1:T})}{q_{\boldsymbol{\phi}}(\boldsymbol{z}_{k,1:T}|\boldsymbol{x}_{1:T}^{\text{obs}})}, \quad \bar{\omega}_k := \frac{\omega_k}{\sum_{\ell=1}^{K}\omega_\ell}. \tag{29}$$

## 4 RELATED WORKS

There are two lines of literature closely related to our method. First line consists of work dealing with the problem of missing data imputation based on Importance Weighted AutoEncoder (IWAE) and another one consists of work designing tailored neural network architecture for supervised learning on multivariate time series data with missing values.

Mattei and Frellsen (2019) proposed MIWAE bound for training DLVMs in the presence of missing data under MAR assumption. Ipsen et al. (2021) modified MIWAE bound suitable to MNAR scenario. Ipsen et al. (2022) extended MIWAE bound to supervised learning task. This line of work

**Table 1:** Classification performances of baseline methods and ours on PhysioNet 2012 dataset.

| Method | AUPRC (↑) | AUROC (↑) | bCE (↓) | bECE (↓) | bBS (↓) | CE (↓) | ECE (↓) | BS (↓) |
|---|---|---|---|---|---|---|---|---|
| GRU-Mean | 0.524 ±0.006 | 0.843 ±0.005 | 0.498 ±0.009 | 0.475 ±0.016 | 0.164 ±0.002 | 0.459 ±0.020 | 0.200 ±0.014 | 0.152 ±0.008 |
| GRU-Simple | 0.426 ±0.012 | 0.810 ±0.008 | 0.621 ±0.062 | 0.400 ±0.042 | 0.201 ±0.017 | 0.450 ±0.027 | 0.138 ±0.033 | 0.147 ±0.011 |
| GRU-Forward | 0.526 ±0.007 | 0.853 ±0.004 | 0.482 ±0.005 | 0.476 ±0.015 | 0.157 ±0.002 | 0.450 ±0.025 | 0.200 ±0.025 | 0.147 ±0.007 |
| GRU-D | 0.553 ±0.004 | 0.863 ±0.001 | 0.474 ±0.010 | 0.473 ±0.034 | **0.152** ±0.004 | 0.434 ±0.040 | 0.184 ±0.033 | 0.140 ±0.015 |
| PhasedLSTM | 0.412 ±0.010 | 0.801 ±0.022 | 0.933 ±0.628 | **0.332** ±0.091 | 0.234 ±0.059 | 0.477 ±0.142 | **0.121** ±0.036 | 0.132 ±0.014 |
| IP-Nets | 0.527 ±0.006 | 0.859 ±0.002 | 0.485 ±0.014 | 0.453 ±0.023 | 0.156 ±0.003 | 0.416 ±0.028 | 0.165 ±0.028 | 0.136 ±0.010 |
| Ours | **0.557** ±0.007 | **0.870** ±0.006 | **0.466** ±0.011 | 0.442 ±0.011 | **0.152** ±0.004 | **0.388** ±0.033 | 0.153 ±0.027 | **0.127** ±0.012 |
|   w/o obsdropout | 0.538 ±0.011 | 0.859 ±0.006 | 0.488 ±0.032 | 0.472 ±0.050 | 0.157 ±0.007 | 0.445 ±0.057 | 0.184 ±0.057 | 0.149 ±0.021 |
|   w/ MLP enc/dec | 0.537 ±0.006 | 0.859 ±0.002 | 0.495 ±0.037 | 0.445 ±0.047 | 0.160 ±0.011 | 0.410 ±0.044 | 0.157 ±0.048 | 0.135 ±0.016 |

**Table 2:** Classification performances of baseline methods and ours on MIMIC-III dataset.

| Method | AUPRC (↑) | AUROC (↑) | bCE (↓) | bECE (↓) | bBS (↓) | CE (↓) | ECE (↓) | BS (↓) |
|---|---|---|---|---|---|---|---|---|
| GRU-Mean | 0.469 ±0.006 | 0.841 ±0.002 | 0.503 ±0.004 | 0.490 ±0.008 | 0.163 ±0.001 | 0.481 ±0.010 | 0.226 ±0.007 | 0.161 ±0.004 |
| GRU-Simple | 0.392 ±0.006 | 0.815 ±0.002 | 0.552 ±0.007 | 0.471 ±0.020 | 0.181 ±0.002 | 0.499 ±0.030 | 0.219 ±0.020 | 0.168 ±0.012 |
| GRU-Forward | **0.519** ±0.005 | 0.857 ±0.002 | 0.471 ±0.003 | 0.497 ±0.011 | **0.155** ±0.001 | 0.468 ±0.018 | 0.230 ±0.016 | 0.154 ±0.006 |
| GRU-D | 0.498 ±0.005 | 0.851 ±0.003 | 0.511 ±0.011 | 0.465 ±0.019 | 0.161 ±0.002 | **0.438** ±0.028 | **0.186** ±0.019 | **0.145** ±0.010 |
| PhasedLSTM | 0.330 ±0.005 | 0.780 ±0.001 | 0.592 ±0.021 | **0.445** ±0.028 | 0.199 ±0.007 | 0.501 ±0.030 | 0.222 ±0.030 | 0.166 ±0.011 |
| IP-Nets | 0.486 ±0.002 | 0.847 ±0.002 | 0.494 ±0.010 | 0.478 ±0.025 | 0.162 ±0.003 | 0.458 ±0.035 | 0.208 ±0.024 | 0.153 ±0.013 |
| Ours | **0.519** ±0.007 | **0.858** ±0.003 | **0.470** ±0.005 | 0.490 ±0.007 | **0.155** ±0.002 | 0.457 ±0.013 | 0.217 ±0.010 | 0.152 ±0.005 |
|   w/o obsdropout | 0.515 ±0.008 | 0.857 ±0.003 | 0.472 ±0.007 | 0.492 ±0.016 | 0.156 ±0.002 | 0.462 ±0.023 | 0.221 ±0.018 | 0.154 ±0.009 |
|   w/ MLP enc/dec | 0.509 ±0.007 | 0.857 ±0.003 | 0.471 ±0.005 | 0.485 ±0.008 | **0.155** ±0.002 | 0.450 ±0.011 | 0.217 ±0.008 | 0.149 ±0.004 |

provides useful framework to train DLVMs under missingness. However, it is not directly applicable for time series data because it cannot model the temporal dependency within a series. There exists previous work to make Deep latent variable models suitable for multivariate time series. For example, Fortuin et al. (2020) proposed VAE architecture which aims to impute multivariate time series data, using Gaussian process prior to encode the temporal correlation in the latent space.

Researchers have developed deep neural network architectures customized to multivariate time series classification task. There have been several architectures showing competitive empirical performance in multivariate time series classification task. Che et al. (2018) modified the architecture of GRU intending to perform supervised learning with sparse covariates by introducing learnable temporal decay mechanism for input and hidden state of GRU. This mechanism has been applied to further research. For example, Cao et al. (2018) employed temporal decay in hidden states of their bidirectional-RNN-based model to capture the missing pattern of irregularly sampled times series. Shukla and Marlin (2019) presented hybrid architecture of interpolation network and classifier. Interpolation network returns fully observed and regularly sampled representation of original time series data. Taking this representation as an input, even common deep neural network model makes good predictive performance.

## 5 EXPERIMENTS

In this section, we demonstarte our method on real-world multivariate time series data with missing values. We compare ours to the baselines on three datasets: PhysioNet 2012 (Silva et al., 2012), MIMIC-III (Johnson et al., 2016) and Human Activity Recognition (Anguita et al., 2013). PhysioNet 2012 and MIMIC-III datasets contain Electronic Health Records of patients from Intensive Care Units (ICU). Human Activity Recognition dataset consists of the 3D coordinate of sensors mounted on the people doing some daily activities such as walking, sitting. See Appendix A for the details of datasets. For all three datasets, we compare classification accuracy and the uncertainty quantification performances. For PhysioNet 2012, we also compare the missing value imputation performance of our methods to the baselines.

For the baselines, we considered GRU classifiers with various imputation methods, and few other deep neural network based methods that are considered to be competitive in the literature. See Appendix A for detailed description of the baselines. For the uncertainty quantification metrics, we compared cross-entropy (CE, equals negative log-likelihood), expected calibration error (ECE), brier score (BS). Especially, for PhysioNet 2012 and MIMIC-III, we also considered balanced versions of them (with "b" in front of the metric names), since those datasets are largely imbalanced so the

**Table 3:** Classification performance of baseline methods and ours on Human Activity Recognition dataset.

| Method | ACC (↑) | CE (↓) | ECE (↓) | BS (↓) |
|---|---|---|---|---|
| GRU-Mean | 0.780 ±0.005 | 0.163 ±0.0121 | 0.019 ±0.006 | 0.046 ±0.002 |
| GRU-Simple | 0.767 ±0.008 | 0.161 ±0.0029 | 0.015 ±0.003 | 0.047 ±0.001 |
| GRU-Forward | **0.798** ±0.007 | 0.152 ±0.0038 | 0.020 ±0.003 | 0.043 ±0.001 |
| GRU-D | 0.789 ±0.004 | 0.150 ±0.0046 | 0.018 ±0.004 | 0.044 ±0.001 |
| Ours | **0.798** ±0.004 | **0.141** ±0.0028 | **0.005** ±0.001 | **0.042** ±0.001 |

**Table 4:** Imputation performance on PhysioNet 2012 dataset.

| Method | MAE (↓) | MRE (↓) |
|---|---|---|
| Mean | 0.696 ±0.001 | 0.994 ±0.000 |
| Forward | 0.399 ±0.001 | 0.574 ±0.003 |
| GRU-D | 0.406 ±0.002 | 0.586 ±0.001 |
| GP-VAE | 0.492 ±0.022 | 0.691 ±0.007 |
| SAITS | 0.694 ±0.001 | 0.995 ±0.000 |
| Ours | **0.391** ±0.002 | **0.564** ±0.002 |
|    w/o supervision | 0.400 ±0.002 | 0.573 ±0.009 |
|    w/o obsdropout | 0.397 ±0.006 | 0.573 ±0.009 |

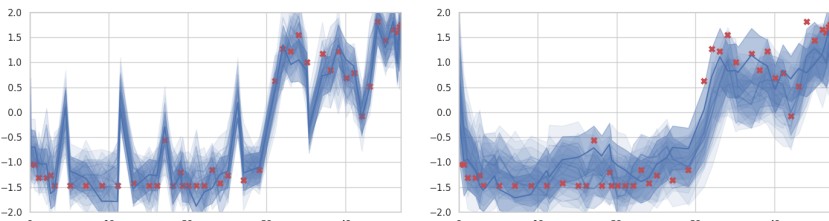

**Figure 2:** Plots of $\boldsymbol{\mu}_{\text{dec}}(\boldsymbol{z}_{1:t}), \boldsymbol{\sigma}^2_{\text{dec}}(\boldsymbol{z}_{1:t})$. (left) Our model which architecture of encoder and decoder is MLP. (right) Our model trained with obsdropout with rate 0.5.

usual uncertainty quantification metrics may be biased. Please refer to Appendix A for the detailed description of the metrics.

## 5.1 CLASSIFICATION RESULTS

We summarize the classification results in Table 1, Table 2, and Table 3. In general, ours acheive the best performance among the competing methods both in terms of prediction accuracy and uncertainty quantification. We also provide an ablation study for our model to see the effect of 1) using time-aware architecture (GRU) for the encoder and decoder of supnotMIWAE, and 2) obsdropout. The results clearly show that both components play important roles for our model. In Appendix B, we provide further results showing the effect of dropout rate $\beta$ for the performance.

## 5.2 IMPUTATION RESULTS

We quantitatively check the imputation performance of our model on PhysioNet 2012 dataset in Table 4, and visually check the imputation quality by changing our model settings in Fig. 2. Although our model is designed for the classification, ours achieved the lowest MAE and MRE, outperforming the baseline (SAITS) specifically designed for the imputation. Especially, the ablation study on the class supervision part $p_{\boldsymbol{\lambda}}(\boldsymbol{y}|\boldsymbol{x}_{1:T})$ and the obsdropout implies that the imputation values generated by our model which was trained to better classify the signals are more "realistic". Fig. 2 highlight the effect of using GRU based encoders and decoders and obsdropout. The values imputed with those techniques form smoother trajectories and better capture the uncertainties in the intervals without observed values.

## 6 CONCLUSION

In this paper, we presented a novel probabilistic framework for multivariate time series classification with missing data. Under the MNAR assumption, we first developed a deep generative model suitable for generating missing values in multivariate time series data. Then we identified an important drawback of the naïve combination of the deep generative models with the classifiers and proposed a novel regularization technique called obsdropout to circumvent that. We demonstrated that ours could classify real-world multivariate time series data more accurately and robustly than existing methods. In this paper, we focused on GRU-based architectures for both generative model and classifier. An interesting future work would be extending our methods with other architectures such as transformers (Vaswani et al., 2017).

**Reproducibility statement**   Please refer to Appendix A for full experimental detail including datasets, models, and evaluation metrics.

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

**Table 5:** Statistics of each dataset.

|                              | PhysioNet2012 | MIMIC-III | Human Activity |
|------------------------------|--------------:|----------:|---------------:|
| Number of samples            | 11,971        | 21,107    | 6554           |
| Number of variables          | 37            | 16        | 12             |
| Maximum number of time steps | 215           | 292       | 221            |
| Mean of number of time steps | 74            | 78        | 162.4          |
| Total missing rate           | 0.843         | 0.655     | 0.923          |

# A  EXPERIMENTAL DETAILS

## A.1  DATASETS

### A.1.1  DATASET DESCRIPTION

**PhysioNet Challenge 2012 (PhysioNet 2012)**   This dataset contains approximately 12,000 Electronic Health Records of adult patients who were admitted to intensive care unit (ICU). Each record contains up to 37 time series variables including vital signs such as heart rate, temperature. All of the variables are measured during the first 48 hours of each patient's admission to ICU and the sampling rate of times series varies among variables. After preprocessing, we have 37 features and 11,971 data point. On this dataset, we conduct missing data imputation task and mortality prediction task, which aims to predict in-hospital mortality of ICU patients using information collected during first 48 hours in ICU.

**MIMIC-III**   MIMIC-III dataset is freely accessible and widely used database which includes de-identified Electronical Health Record of patients who stayed in ICU of Beth Israel Deaconess Medical Center from 2001 to 2012. It originally consists of approximately 57,000 records of patients stayed in ICU. Records including various variables such as medications, in-hospital mortality and vital signs. Harutyunyan et al. (2019) set variety of benchmark tasks based on subset of this database. Among them, we conduct binary in-hospital mortality prediction task. After preprocessing, our dataset contains 16 features and 21,107 data points. For this dataset, we conduct mortality prediction task which is identical to PhysioNet2012 classification task.

**UCI Localization Data for Person Activity (UCI Human Activity)**   This dataset includes records of five people doing some usual activities such as walking or sitting. All people wear sensors on their right ankle,left,belt and chest. During activities, and the sensors record their position in the form of three-dimensional coordinates at very short intervals. Activities of each people at certain time point classified into one of 11 classes and recorded with the position of sensors. After preprocessing, we have total 6554 time series with 12 features(3-dimensional coordinates of 4 devices). Using this preprocessed data, we conduct online-prediction task. The objective of this task is to classify each individual's activity per time point based on the position of sensors.

### A.1.2  DATA PREPROCESSING

For all dataset, we basically standardize the numerical covariates so that all features have zero mean and unit variance, respectively.

**PhysioNet2012 and MIMIC-III**   Since there is no fixed rule for preprocessing Physionet2012 and MIMIC-III database, researchers usually preprocess the raw data on their own so that there are countless possibilities for the form of preprocessed dataset. Therefore, it is difficult for practitioners to compare experimental results with other works. For the comparability, we employ python package medical-ts-datasets (Horn et al., 2020) which provides the unified data preprocessing pipeline for Physionet2012 and MIMIC-III datasets. For both dataset, patients who have more than 1000 time steps or have no observed time series data were excluded from the dataset. Also, discretizing the time step of data by hour and aggregate the measurement is frequently used to preprocess Physionet2012 in previous work (Rubanova et al., 2019), but this package preserves much more original time series variables while preprocessing than hourly based aggregation preprocessing for both dataset. We follow the preprocessing of the `medical_ts_datasets`[1] library.

---

[1] https://github.com/ExpectationMax/medical_ts_datasets

**Table 6:** The number of parameters of baseline models and our model on each dataset.

| Method | PhysioNet 2012 | MIMIC-III | Human Activity |
|---|---|---|---|
| GRU-Mean | 226,817 | 210,689 | 84,647 |
| GRU-Simple | 351,745 | 301,569 | 85,117 |
| GRU-Forward | 294,145 | 276,225 | 85,687 |
| GRU-D | 332,363 | 292,897 | 86,566 |
| PhasedLSTM | 511,745 | 445,441 | - |
| IP-Nets | 353,151 | 301,841 | - |
| Ours | 262,980 | 224,305 | 84,251 |

**UCI Human Activity**   For comparability, we decide to preprocess this dataset based on (Rubanova et al., 2019). However, we modify some part of dataset to apply to our code implementation. We standardize the dataset while (Rubanova et al., 2019) did not standardize the dataset. Also, we make new variable that records the time index of lastly observed value of each data point to distinguish between missingness and meaningless padding.

## A.2  DETAILS FOR CLASSIFICATION EXPERIMENTS

For all experiments, we use five different seeds to conduct experiments.

### A.2.1  BASELINE METHODS

- **GRU-mean**: Missing value is simply replaced with the empirical mean of each variable.

- **GRU-forward**: Missing entries are filled with previously observed value.

- **GRU-simple**: concatenate the mask $s_t$, and the *time-interval* $\delta_t$ along with the imputed vector $\hat{x}_t$. The concatenated vector $[\hat{x}_t, s_t, \delta_t]$ is then fed into GRU.

- **GRU-D**: Missing values are imputed as a weighted mean of the last observed $x_{t',j}$ and the mean $\bar{x}_j$ with the learnable weight.

- **Phased-LSTM**: This model is LSTM variants designed to deal with long sequence input by introducing time gate in their cell to prevent memory decay when useful information is absence for a long time.

- **Interpolation-Prediction Network(IP-Nets)**: Instead of directly imputing missing values, this model employed semi-parametric interpolation network that makes regularly spaced representation of irregularly sampled time series data. Then, this representation is fed into prediction network such as GRU.

Since we conduct online-prediction task on Human Activity dataset, we do not consider IP-Nets as baseline models because this model use future information when conducting interpolation.

### A.2.2  TRAINING DETAILS

In order to conduct experiments fairly, we fix the parameters of every model similar, or at least set our model to have relatively small number of parameters compared to other baseline. Also, we use Adam optimizer with learning rate 0.0001 and batch size 128 for all models for MIMIC-III in-hospital mortality prediction task. In Physionet2012 experiment, we use Adam optimizer with learning rate 0.001 and batch size 128 for all models. For online-prediction task, we also employ Adam optimizer with learning rate 0.001 and batch size 128.

We employ early stopping for all classification experiments. For the mortality prediction tasks, we set early stopping patience to 10 epochs and set the area under the precision-recall curve (AUPRC) of the validation data as the early stopping criterion. For Human activity prediction task, we used validation accuracy as stopping criterion and set early stopping patience to 20 epochs. Since the label imbalance of Physionet2012 and MIMIC-III is extreme, we oversample the mortality class to train models on the balanced batches.

**Table 7:** Hyperparameter settings for classification experiments

| Dataset | Method | Hyperparameters |
|---|---|---|
| PhysioNet2012 | GRU-Mean | n_units: 256, dropout: 0.0, recurrent_dropout: 0.0 |
| | GRU-Simple | n_units: 256, dropout: 0.0, recurrent_dropout: 0.0 |
| | GRU-Forward | n_units: 256, dropout: 0.0, recurrent_dropout: 0.0 |
| | GRU-D | n_units: 256, dropout: 0.0, recurrent_dropout: 0.0 |
| | PhasedLSTM | n_units: 256, use_peepholes: False, leak: 0.01,period_init_max: 1000.0 |
| | IP-Nets | n_units: 256, imputation_stepsize: 1 , reconst_fraction: 0.5 |
| | Ours | n_train_latents: 10, n_train_samples: 1, n_test_latents: 20, n_test_samples: 30, n_hidden: 128, z_dim: 32, n_units: 128, observe_dropout: 0.5 |
| MIMIC-III | GRU-Mean | n_units: 256, dropout: 0.0, recurrent_dropout: 0.0 |
| | GRU-Simple | n_units: 256, dropout: 0.0, recurrent_dropout: 0.0 |
| | GRU-Forward | n_units: 256, dropout: 0.0, recurrent_dropout: 0.0 |
| | GRU-D | n_units: 256, dropout: 0.0, recurrent_dropout: 0.0 |
| | PhasedLSTM | n_units: 256, use_peepholes: False, leak: 0.01, period_init_max: 1000.0 |
| | IP-Nets | n_units: 256, imputation_stepsize: 1, reconst_fraction: 0.5 |
| | Ours | n_train_latents: 10, n_train_samples: 1, n_test_latents: 20, n_test_samples: 30, n_hidden: 128, z_dim: 32, n_units: 128, observe_dropout: 0.1 |
| Human Activity | GRU-Mean | n_units: 160, dropout: 0.0, recurrent_dropout: 0.0 |
| | GRU-Simple | n_units: 132 , dropout: 0.0, recurrent_dropout: 0.0 |
| | GRU-Forward | n_units: 140 , dropout: 0.0, recurrent_dropout: 0.0 |
| | GRU-D | n_units: 135 , dropout: 0.0, recurrent_dropout: 0.0 |
| | Ours | n_train_latents: 10, n_train_samples: 1, n_test_latents: 10, n_test_samples: 10, n_hidden: 128, z_dim: 32, n_units: 128, observe_dropout: 0.2 |

### A.2.3 HYPERPARAMETERS

See Table 6 and Table 7 for hyperparameter settings of our model and baseline methods for all classification experiments.

### A.2.4 EVALUATION METRICS

For the classification task, we evaluate all models in terms of both predictive accuracy and predictive uncertainty. We use the area under precision recall curve (AUPRC), the area under receiver operating characteristic (AUROC), and the accuracy (ACC) to evaluate the predictive performance. To measure the uncertainty calibration of the model, we use cross entropy (CE), expected calibration error (ECE) and brier score (BS) for comparing calibration. In addition, we also check the balanced versions of uncertainty metrics due to severe class imbalance of datasets.

**Balanced metric**  In supervised dataset $\mathcal{D}$, which contains input data $x$ and a corresponding label $y$, we simply re-weight each uncertainty metric by class ratio.

$$M_{\text{bal}}(\mathcal{D}) = \frac{1}{C} \sum_c M(\mathcal{D}_c) \tag{30}$$

Here, $\mathcal{D}_c$ is a subset of the dataset $\mathcal{D}$ which only contains the label $y = c$.

**Accuracy Metrics**  Accuracy metrics are defined using the following terms, where $tp$, $tn$, $fn$, and $fp$ denote true positive, true negative, false negative, and false positive respectively.

$$\text{accuracy} = \frac{tp + tn}{tp + fp + fn + tn} \tag{31}$$

$$\text{precision} = \frac{tp}{tp + fp} \tag{32}$$

$$\text{recall} = \frac{tp}{tp + fn} \tag{33}$$

$$\text{sensitivity} = \frac{tp}{tp + fn} \tag{34}$$

**Table 8:** Hyperparameter settings for imputation experiments

| Dataset | Method | Hyperparameters |
|---------|--------|-----------------|
| PhysioNet2012 | Mean | - |
| | Forward | - |
| | GRU-D | n_units: 256, dropout: 0.0, recurrent_dropout: 0.0 |
| | Ours w/ obsdropout | n_train_latents: 10, n_train_samples: 1, n_test_latents: 20, n_test_samples: 30, n_hidden: 128, z_dim: 32, n_units: 128, observe_dropout: 0.5 |
| | Ours w/o supervision | n_train_latents: 10, n_train_samples: 1, n_test_latents: 20, n_test_samples: 30, n_hidden: 128, z_dim: 32, n_units: 128, |
| | Ours w/o obsdropout | n_train_latents: 10, n_train_samples: 1, n_test_latents: 20, n_test_samples: 30, n_hidden: 128, z_dim: 32, n_units: 128, observe_dropout: 0.0 |
| | GP-VAE | n_hidden: 128, z_dim: 35, length_scale: 7, kernel_scale: 1, n_latents: 1,beta: 0.2 |
| | SAITS | n_groups:5 , n_head:1 , n_group_inner_layers:1 , d_model:258,d_inner:128,d_k:64,d_v:64,MIT:True, input_with_mask: True, MIT_missing_rate:0.2 , n_units:128, n_hidden:128 |

- **AUPRC** (Schütze et al., 2008): area under precision recall curve.

- **AUROC**: area under receiver operating characteristic, area under sensitivity curve.

## A.3 DETAILS FOR IMPUTATION EXPERIMENTS

Basically, we perform imputation experiments on the testdataset of Physionet2012. We randomly delete 10% of observed data for testing imputation performance of models. We measure the performance for five different seeds.

### A.3.1 BASELINE METHODS

- **Mean**: Replace missing values with global mean.

- **Forward**: Impute missing value with previously observed value

- **GRU-D**: Missing values are imputed as a weighted mean of the last observed $x_{t',j}$ and the mean $\bar{x}_j$ with the learnable weight.

- **GP-VAE**: This model is VAE-based probabilistic imputation method proposed by Fortuin et al. (2020). This method employ GP-prior to encode the temporal correlation in the latent space.

- **SAITS**: This model is self-attention based imputation model which Du et al. (2022) proposed.

- **Ours w/o supervision**: To analyze the effect of supervised signal to the imputation performance, we just remove supervised term from our training objective and train only generative part of our architecture.

- **Ours w/o dropout**: Our model without obsdropout.

### A.3.2 EVALUATION METRICS

For the imputation task, we evaluate all methods in terms of MRE(Mean Relative Error) and MAE(Mean Absolute Error).

### A.3.3 TRAINING DETAILS

For GRU-D and Our model, we apply early stopping rule identical to mortality prediction task since imputation performance of our model is deeply related to classification error. For GP-VAE,SAITS and Our model w/o supervised signals, we do not use early stopping and train for 50 epochs for each model.

### A.3.4 HYPERPARAMETERS

**Table 9:** Effect of obsdropout rate for ours on PhysioNet 2012 dataset.

| Dropout | AUPRC (↑) | AUROC (↑) | bCE (↓) | bECE (↓) | bBS (↓) | CE (↓) | ECE (↓) | BS (↓) |
|---|---|---|---|---|---|---|---|---|
| 0.0 | 0.538 ±0.011 | 0.859 ±0.006 | 0.488 ±0.032 | 0.472 ±0.050 | 0.157 ±0.007 | 0.445 ±0.057 | 0.184 ±0.057 | 0.149 ±0.021 |
| 0.1 | 0.545 ±0.014 | 0.862 ±0.007 | 0.473 ±0.023 | 0.474 ±0.023 | 0.153 ±0.006 | 0.437 ±0.023 | 0.184 ±0.029 | 0.146 ±0.008 |
| 0.2 | 0.546 ±0.005 | 0.865 ±0.004 | 0.480 ±0.025 | 0.457 ±0.033 | 0.154 ±0.006 | 0.416 ±0.037 | 0.161 ±0.034 | 0.138 ±0.014 |
| 0.3 | 0.554 ±0.008 | 0.868 ±0.001 | 0.464 ±0.006 | 0.477 ±0.039 | **0.149** ±0.002 | 0.434 ±0.056 | 0.181 ±0.040 | 0.145 ±0.021 |
| 0.4 | 0.558 ±0.005 | 0.868 ±0.006 | 0.470 ±0.014 | **0.452** ±0.026 | 0.152 ±0.004 | 0.401 ±0.035 | **0.155** ±0.027 | **0.132** ±0.013 |
| 0.5 | **0.561** ±0.003 | **0.871** ±0.004 | 0.462 ±0.016 | 0.456 ±0.021 | **0.149** ±0.005 | **0.400** ±0.025 | 0.158 ±0.020 | **0.132** ±0.009 |
| 0.6 | 0.556 ±0.005 | 0.869 ±0.002 | **0.458** ±0.008 | 0.474 ±0.021 | **0.149** ±0.002 | 0.425 ±0.027 | 0.179 ±0.018 | 0.141 ±0.011 |
| 0.7 | 0.554 ±0.014 | 0.867 ±0.003 | 0.462 ±0.013 | 0.471 ±0.026 | 0.151 ±0.004 | 0.425 ±0.029 | 0.179 ±0.025 | 0.140 ±0.011 |
| 0.8 | 0.558 ±0.010 | 0.864 ±0.009 | 0.472 ±0.031 | 0.489 ±0.033 | 0.153 ±0.008 | 0.455 ±0.032 | 0.194 ±0.033 | 0.152 ±0.012 |
| 0.9 | 0.546 ±0.010 | 0.859 ±0.006 | 0.469 ±0.015 | 0.486 ±0.016 | 0.154 ±0.003 | 0.453 ±0.023 | 0.198 ±0.020 | 0.152 ±0.009 |

**Table 10:** Effect of obsdropout rate for ours on MIMIC-III dataset.

| Dropout | AUPRC (↑) | AUROC (↑) | bCE (↓) | bECE (↓) | bBS (↓) | CE (↓) | ECE (↓) | BS (↓) |
|---|---|---|---|---|---|---|---|---|
| 0.0 | 0.515 ±0.008 | 0.857 ±0.003 | 0.472 ±0.007 | 0.492 ±0.016 | 0.156 ±0.002 | 0.462 ±0.023 | 0.221 ±0.018 | 0.154 ±0.009 |
| 0.1 | **0.519** ±0.007 | 0.858 ±0.003 | 0.470 ±0.005 | 0.490 ±0.007 | **0.155** ±0.002 | **0.457** ±0.013 | **0.217** ±0.010 | **0.152** ±0.005 |
| 0.2 | **0.519** ±0.006 | **0.859** ±0.003 | 0.468 ±0.004 | 0.501 ±0.009 | **0.155** ±0.002 | 0.472 ±0.014 | 0.228 ±0.009 | 0.158 ±0.005 |
| 0.3 | 0.518 ±0.004 | **0.859** ±0.002 | **0.469** ±0.002 | 0.494 ±0.008 | **0.155** ±0.001 | 0.462 ±0.013 | 0.220 ±0.009 | 0.154 ±0.005 |
| 0.4 | 0.518 ±0.004 | 0.858 ±0.003 | 0.470 ±0.003 | 0.493 ±0.015 | **0.155** ±0.001 | 0.461 ±0.025 | 0.220 ±0.017 | 0.153 ±0.009 |
| 0.5 | 0.511 ±0.010 | 0.855 ±0.003 | 0.475 ±0.006 | **0.488** ±0.010 | 0.157 ±0.002 | **0.457** ±0.013 | **0.217** ±0.013 | **0.152** ±0.005 |
| 0.6 | 0.504 ±0.010 | 0.851 ±0.003 | 0.483 ±0.005 | 0.501 ±0.011 | 0.160 ±0.002 | 0.488 ±0.015 | 0.229 ±0.010 | 0.163 ±0.006 |
| 0.7 | 0.480 ±0.036 | 0.840 ±0.017 | 0.499 ±0.020 | 0.497 ±0.010 | 0.166 ±0.007 | 0.495 ±0.020 | 0.236 ±0.024 | 0.163 ±0.006 |
| 0.8 | 0.461 ±0.032 | 0.831 ±0.015 | 0.515 ±0.018 | 0.490 ±0.006 | 0.171 ±0.006 | 0.498 ±0.020 | 0.235 ±0.029 | 0.163 ±0.006 |
| 0.9 | 0.419 ±0.024 | 0.810 ±0.010 | 0.543 ±0.011 | 0.496 ±0.029 | 0.181 ±0.004 | 0.534 ±0.050 | 0.268 ±0.030 | 0.174 ±0.019 |

See Table 8 for detailed hyperparameter settings for imputation experiment.

# B   ADDITIONAL EXPERIMENTS

We conduct numerous ablation experiments to analyze the effect of obsdropout method. In PhysioNet2012 dataset, our models with obsdropout method outperforms at all rates. Also, in MIMIC-III datset, our technique works well on reasonable rates. Although there is much room for further analysis of effect of obsdropout on the predictive performance, these results are sufficient to show that our technique is generally effective.

