# OpenReview forum: "Probabilistic Imputation for Time-series Classification with Missing Data"
_ICLR.cc/2023/Conference — Submitted to ICLR 2023_

### Official Review · Reviewer_wRvk · 2022-10-20

**Confidence:** 3
**Correctness:** 4
**Technical Novelty And Significance:** 3
**Empirical Novelty And Significance:** 3
**Recommendation:** 5

**Clarity, Quality, Novelty And Reproducibility:**

The paper is generally well written. The presentation is clear.
Accounting for MNAR for latent state-space models is novel, as far as I am aware.

**Strength And Weaknesses:**

Positives:
- The paper proposes a probabilistic model for time series data that allows for very flexible assumptions for the missing values. Imputation for the time series setting/for non-linear latent state-space models has not been explored in the literature, as far as I am aware.
- The method appears to yield better empirical performance on two benchmark datasets, compared to imputation approaches using GRUs.

Negatives:
-There appears to be a lack of empirical comparison to other probabilistic imputation approaches, such as Fortuin et al., 2020. Even if such approaches make strong assumptions on the missingness patterns, it would be insightful to see how this translates into the empirical forecasting performance.
- The missingness model appears relatively simple compared to the other components (encoder, classifier) in that it does not depend on the full history of the observations via the RNN latent state. A more flexible model should not require 'data-specific design' choices.

Comments/Actionable feedback:
-For the suggested bound with the dropout variables in (22), does this constitute a lower bound on something sensible?
- My feeling is that even without dropout, there should be an incentive to learn non-trivial imputations, because I am not only minimising the classification error, but also the log-probability of the observed time series.
- A semi-supervised approach to regularise the missing data has been suggested in Sahra Ghalebikesabi, Rob Cornish, Chris Holmes, Luke Kelly: Deep Generative Missingness Pattern-Set Mixture Models. AISTATS 2021. I was wondering how this relates to the dropout procedure suggested here?





**Summary Of The Paper:**

The authors suggest a sequential VAE/IWAE framework for imputation and classification of time series data that is particularly applicable
 under the missing not at random (MNAR) setting. To improve the imputation performance, the authors suggest to mask some observed values and subsequently reconstruct them. The method is compared agains imputation approaches using GRUs that are not fully probabilistic.


---
Update after the authors' response:
The authors have added an additional probabilistic imputation benchmark model which has been found to be outperformed by the approach suggested here. This improves the empirical validation of the new approach.

---

**Summary Of The Review:**

The proposed modelling setup is new and addresses a relevant question that has not yet been addressed previously. The model (encoder, decoder) is somewhat standard. Including the missingness model is new, as well as the dropout procedure, along with the variational bound. The empirical performance appears favourable compared to imputations with GRUs, but in my opinion, a more detailed empirical comparison with other probabilistic imputation approaches would improve the paper.

---

> ### Author Response · Authors · 2022-11-12
> **Response to Reviewer wRvK**
>
> Thanks for recognizing our technical novelty and significance. We’d like to answer your comments and feedback.
>
> > There appears to be a lack of empirical comparison to other probabilistic imputation approaches, such as Fortuin et al., 2020. Even if such approaches make strong assumptions on the missingness patterns, it would be insightful to see how this translates into the empirical forecasting performance.
>
> We evaluated the imputation performance of GP-VAE (Fortuin et al., 2020) and updated Table 4 in our paper. Our method outperforms GP-VAE in terms of both MAE and MRE.
>
> > The missingness model appears relatively simple compared to the other components (encoder, classifier) in that it does not depend on the full history of the observations via the RNN latent state. A more flexible model should not require 'data-specific design' choices.
>
> It makes sense, but at least in our setting, classification performance of our model does not change even when we change the missingness model to RNN as shown in Table A1. We expect that data-specific design of missingness models is necessary to get better performance without obsdropout.
>
> Table A1. Classification performance of our method using different missingness models on PhysioNet 2012 dataset.
> ​​Missingness Model | AUPRC (↑)     | AUROC (↑)     | CE (↓)        | ECE (↓)       | BS (↓)
> ------------------|---------------|---------------|---------------|---------------|--------------
> MLP (Table 1)     | 0.557 ± 0.007 | 0.870 ± 0.006 | 0.388 ± 0.033 | 0.153 ± 0.027 | 0.127 ± 0.012
> GRU               | 0.556 ± 0.009 | 0.870 ± 0.003 | 0.406 ± 0.041 | 0.161 ± 0.030 | 0.134 ± 0.015
>
> > For the suggested bound with the dropout variables in (22), does this constitute a lower bound on something sensible?
>
> We added an explanation for the suggested bound to page 7 in the paper for better understanding (eq 23-27). This bound constitutes a sensible lower bound.
>
> > My feeling is that even without dropout, there should be an incentive to learn non-trivial imputations, because I am not only minimising the classification error, but also the log-probability of the observed time series.
>
> The ablation in Table 4 (ours w/o supervision) shows that ours trained without classification loss performs better than the baselines, indicating that the log-probability term is contributing to better imputations to some extent. Still, in principle, please note that the model can still cheat the log-probability loss for observed variables by 1) encoding only the information of observed values into latent, and 2) producing non-sense values for imputations (i.e., it is essentially autoencoding only the observed variables without taking missing imputations into account). By doing this the model would still be able to maximize the log-probability of the observed values. Note also that ours with classification loss + obsdropout is significantly improving the imputation performance on the ones without them.
>
> > A semi-supervised approach to regularise the missing data has been suggested in Sahra Ghalebikesabi, Rob Cornish, Chris Holmes, Luke Kelly: Deep Generative Missingness Pattern-Set Mixture Models. AISTATS 2021. I was wondering how this relates to the dropout procedure suggested here?
>
> PSMVAE (Ghalebikesabi et al. 2021) utilizes a regularizer called a probabilistic semi-supervision. Both the probabilistic semi-supervision and obsdropout are methods to compensate for the limitation of training incomplete data. However, obsdropout randomly discards some portion of the observed part of data and fills the missing value generated by the decoder, whereas the probabilistic semi-supervision generates x from the decoder and trains the model so that the observed part of x is close to the observed part of data and the missing part of x is close to the missing values generated from the encoder (the encoder of PSMVAE generates not only the latent variable z but also missing values). In terms of their effects, obsdropout makes the prediction task more focused on imputation, but the probabilistic semi-supervision prevents the generated missing values to overfit to the distribution of the observed data.
>
> Reference
>
> - V. Fortuin, D. Baranchuk, G. Raetsch, and S. Mandt. GP-VAE: Deep probabilistic time series imputation. In Proceedings of The 23rd International Conference on Artificial Intelligence and Statistics (AISTATS 2020), 2020. (ArXiv link [https://arxiv.org/abs/1907.04155](https://arxiv.org/abs/1907.04155))
> - S. Ghalebikesabi, R. Cornish, C. Holmes, L. Kelly. Deep Generative Missingness Pattern-Set Mixture Models. In Proceedings of The 24th International Conference on Artificial Intelligence and Statistics (AISTATS 2021), 2021. (ArXiv link [https://arxiv.org/abs/2103.03532](https://arxiv.org/abs/2103.03532))

---

### Official Review · Reviewer_ssPS · 2022-10-24

**Confidence:** 3
**Clarity, Quality, Novelty And Reproducibility:** Good
**Correctness:** 3
**Technical Novelty And Significance:** 4
**Empirical Novelty And Significance:** 3
**Recommendation:** 5

**Strength And Weaknesses:**

Pros:
1. This paper is probabilistic and uncertainty is considered for classification
2. The regularization techniques makes sense

Cons:
1. Several recent works have shown that GRU-D is not good at handle irregularly sampled data compared to ODE based methods, is it possible to extent this paper to handle irregularly-sampled data well?
2. In Table1 and Table2, the improvement over GRU-D is marginal, so I am not sure this method is as effective as claimed.



**Summary Of The Paper:**

THis paper tackles time series classification problems with missing data. A deep generative model is used to impute missing data first before classification. The classifier is trained to capture the predictive uncertainty due to the multiple possibilities of imputations. A novel regularization technique is proposed that can promote the model to produce useful imputation values that actually
help classifcation.

**Summary Of The Review:**

Can't handle irregularly sampled data well, improvement is marginal

---

> ### Author Response · Authors · 2022-11-12
> **Response to Reviewer ssPS**
>
> Thanks for acknowledging the novelty of our regularizer. We would like to answer your comments.
>
> > Several recent works have shown that GRU-D is not good at handle irregularly sampled data compared to ODE based methods, is it possible to extent this paper to handle irregularly-sampled data well?
>
> Thanks for your question that could potentially lead to the improvement of our method. The architecture of the encoder-decoder (GRU-D and GRU) we employed in this paper can be modified since our framework is not dependent on the architecture of the neural network model. For example, Latent ODE (Rubanova et al., 2019) also employs encoder-decoder structure and variational inference, this model is also compatible with our framework. Therefore, we can adapt the Latent ODE model in our framework. Since the obsdropout only replaces some observed values with generated values before being fed into the classifier, we can use it without modification with the ODE based methods. We expect future work that incorporates ODE based method and our framework to better deal with irregularly sampled time series data.
>
> > In Table1 and Table2, the improvement over GRU-D is marginal, so I am not sure this method is as effective as claimed.
>
> Our method outperforms GRU-D with a wide margin in terms of AUROC considering the standard deviation of results. You may think of our results as trivial improvement, but Latent ODE (Rubanova et al., 2019) shows similar AUROC with GRU-D in PhysioNet 2012 mortality prediction task (Table 6 of this paper). This shows that even ODE based models have difficulty outperforming GRU-D in spite of the good ability they have in handling irregularly sampled time series.
>
> Reference
> - Y. Rubanova, R. T. Q. Chen, and D. Duvenaud. Latent ODEs for Irregularly-Sampled Time Series. In Advances in Neural Information Processing Systems 32 (NeurIPS 2019), 2019. (ArXiv link [https://arxiv.org/abs/1907.03907](https://arxiv.org/abs/1907.03907))

---

### Official Review · Reviewer_For5 · 2022-10-24

**Confidence:** 3
**Correctness:** 3
**Technical Novelty And Significance:** 3
**Empirical Novelty And Significance:** 3
**Recommendation:** 5

**Clarity, Quality, Novelty And Reproducibility:**

The paper is overall organized and easy to follow. Regarding technical novelty, this paper presents some non-trivial extensions to existing work, but some assumptions (like driving the imputation using prediction only) might need stronger justifications. The detailed model specification, hyperparameter setting, and training details are lacking, making the paper less reproducible.

**Strength And Weaknesses:**

Strengths:
- The proposed model is reasonable and sound. Detailed analysis of the problem as well as existing work is provided.
- Although the proposed supnotMIWAE is largely a variant of supMIWAE, it adds new elements to tackle the MNAR and prediction and the formulation is overall sound.
- Overall, the paper is organized and easy to follow.

Weaknesses:
- My main concern regarding the method is that the prediction task may not be enough to drive the learning of imputation, especially MNAR. Even with the obsDropout regularization incorporated, the model only generates imputation that helps most the prediction but may not be the most accurate imputation.
- The empirical evaluation, compared against existing methods like GRU-forward and GRU-D, seems weak. The improvement over the SOTA baselines is quite marginal.

**Summary Of The Paper:**

This paper focuses on imputation and prediction for time series with missing data. It extends an existing deep generative model to impute the missing values in the time series and make predictions given both the observations and imputations. A regularization method "obsDropout" is proposed to randomly drop some observed entries to force the prediction model to focus more on the imputed entries to drive the generation of good imputations.

**Summary Of The Review:**

Overall, this paper presents some interesting extensions of the existing deep generative model for time series missing data imputation. The formulations are sound. However, some methodological design needs stronger justification and the empirical evaluation seems weak.

---

> ### Author Response · Authors · 2022-11-12
> **Response to Reviewer For5**
>
> Thanks for evaluating our work as interesting. We want to elaborate more on the justification related to your concerns.
>
> > My main concern regarding the method is that the prediction task may not be enough to drive the learning of imputation, especially MNAR. Even with the obsDropout regularization incorporated, the model only generates imputation that helps most the prediction but may not be the most accurate imputation.
>
> Since there is no ground truth for missing data in the datasets we used to train our model (and this is common for most of the real-world datasets), it is difficult to define “what a good imputation is” in the first place. Please note that, however, our ultimate goal of the imputation is to “better classify” the time-series; in other words, unless there are strong inductive biases or even ground truths for missing data, in our opinion, encouraging the model to impute the missings in a way to enhance classification performance is the best option we can choose. Moreover, as shown in Table 4, where we deliberately drop 10% of the observed values and compare the imputation performances, our method clearly benefits from the supervised signal; ours with supervised loss is clearly better than ours without it, indicating that the classification loss is actually helping the model to impute values closer to the true values.
>
> > The empirical evaluation, compared against existing methods like GRU-forward and GRU-D, seems weak. The improvement over the SOTA baselines is quite marginal.
>
> We respectfully disagree with this. Ours perform significantly better than baselines (in terms of accuracy or uncertainty calibration) for most of the datasets we tested, except for MIMIC-III where the improvements were marginal. We believe that this is mainly due to the characteristic of the MIMIC-III dataset; for instance, GRU-Forward works well for this particular dataset because there are not many drastic transitions in the dataset so it is generally better to make predictions based on the observations from the preceding time steps. However, as shown in the results for the other datasets, GRU-Forward would not succeed for the datasets not exhibiting such behavior. On the other hand, ours perform better than baselines regardless of the datasets we tested, indicating that ours has the flexibility to adapt to the characteristic of the datasets.
>
> > The detailed model specification, hyperparameter setting, and training details are lacking, making the paper less reproducible.
>
> Thank you for pointing out. We added detailed specification of model, hyperparameters, and training details in the appendix.

---

### Official Review · Reviewer_1NJg · 2022-10-25

**Confidence:** 4
**Correctness:** 4
**Technical Novelty And Significance:** 3
**Empirical Novelty And Significance:** 4
**Recommendation:** 8

**Clarity, Quality, Novelty And Reproducibility:**

Clarity:  The paper is well-written and clear. The method and model are well-motivated and clear

Quality: The derivations are technically correct (to the best of my knowledge). The modeling choices are appropriate for the different “modules” and their purpose (missingness, time series, classifier). The empirical validation is convincing

Novelty: Somewhat incremental.

Reproducibility: The paper is accompanied by code for the model and experiments (which I did not go through)



**Strength And Weaknesses:**

## Strength
* The modeling choices and inference method is well motivated and principled
* The problem of imputation with NMAR in the multivariate time-series setup with a hybrid deep generative model is novel and potentially impactful
* Empirical results demonstrate the strength of the proposal

## Weakness:
* The methodology is incremental

typos:
Irrelvent -> irrelevant (page 3)


**Summary Of The Paper:**

This paper introduces a deep generative model for multivariate time series with missing data (in the most general setup of missing-not-at-random, MNAR). The development of the model follows the same core design principles presented in [1], [2], and [3], extending it and proposing a joint hybrid model for the time series, the missing mechanism, and a classifier. The inference of such a model can be intractable in general, thus an approximation for the log-likelihood is proposed, adapting the Importance Weighted AutoEncoder (IWAE) lower bound for the proposed model (which defines the loss function to be optimized). Furthermore, it is proposed a regularization for this lower bound by dropping more portions of the observations, encouraging imputations that contribute towards the classification tasks. The empirical validation is done using multiple datasets and comparing with multiple baselines and state-of-the-art.

[1] P. Mattei and J. Frellsen. MIWAE: deep generative modelling and imputation of incomplete data sets. ICML 2019.
[2] N. B. Ipsen, P. Mattei, and J. Frellsen. not-MIWAE: deep generative modelling with missing not at random data. ICLR 2021.
[3] N. B. Ipsen, P. Mattei, and J. Frellsen. How to deal with missing data in supervised deep learning?. ICLR 2022.




**Summary Of The Review:**

Overall, this a solid and well-motivated work, tackling an relevant problem (NMAR imputation for time-series). My only concern is the limited technical novelty. That being said, I recommend acceptance.

---

> ### Author Response · Authors · 2022-11-12
> **Response to Reviewer 1NJg**
>
> Thanks for your positive view supporting our novel trial in the imputation for the multivariate time series in the MNAR setting. Our answer for your concern is as follows:
>
> > The method is incremental.
>
> First of all, we introduce a novel probabilistic framework combining deep generative models for sequential data, MNAR misingness model, and classifier. We further propose an effective variational inference algorithm based on importance sampling. Moreover, the idea of exploiting the dropout for the observed values to enhance missing imputation has never been explored before to the best of our knowledge. The individual components may not seem novel, but the combination of those ingredients in the context of time-series classification with missing values is novel.

---

### Author Response · Authors · 2022-11-18
**Dear All Reviewers,**

We appreciate for your constructive comments. We'd like to ask for going over our responses and the revision before the end of the author-reviewer discussion. We have faithfully responded to your comments and reflected them in the revision, so we'd like to hear back from you whether they resolve your concerns or not.

Thanks, Authors.

---

### Decision · Program_Chairs · 2023-01-20

**Decision:**

Reject

**Justification For Why Not Higher Score:**

Marginal novelty.

**Justification For Why Not Lower Score:**

n/a

**Metareview: Summary, Strengths And Weaknesses:**

This interesting paper has been thoroughly reviewed by four knowledgeable reviewers. Three of them assessed it below the threshold for acceptance, one suggested an accept. Even the strongest review points out the key deficiency of the paper in its current form: its limited novelty versus the prior work. Others point the novelty as marginal, and suggest that more comprehensive empirical studies would need to be conducted to reinforce the claims being made. In summary, I find this work interesting and very promising, but at its current stage it is below the threshold of ICLR acceptance. However, I would like to encourage the authors to continue maturing it.